# Absolute Configuration of Mycosporine-Like Amino Acids, Their Wound Healing Properties and In Vitro Anti-Aging Effects

**DOI:** 10.3390/md18010035

**Published:** 2019-12-31

**Authors:** Maria Orfanoudaki, Anja Hartmann, Mostafa Alilou, Thomas Gelbrich, Patricia Planchenault, Séverine Derbré, Andreas Schinkovitz, Pascal Richomme, Andreas Hensel, Markus Ganzera

**Affiliations:** 1Institute of Pharmacy, Pharmacognosy, University of Innsbruck, Innrain 80-82, 6020 Innsbruck, Austria; Maria.Orfanoudaki@uibk.ac.at (M.O.); mostafa.alilou@student.uibk.ac.at (M.A.); markus.ganzera@uibk.ac.at (M.G.); 2Institute of Pharmacy, Pharmaceutical Technology, University of Innsbruck, Innrain 52c, 6020 Innsbruck, Austria; thomas.gelbrich@uibk.ac.at; 3SONAS, EA921, University of Angers, SFR QUASAV, Faculty of Health Sciences, Department of Pharmacy, 16 Bd Daviers, 49045 Angers, France; patricia.planchenault@univ-angers.fr (P.P.); severine.derbre@univ-angers.fr (S.D.); andreas.schinkovitz@univ-angers.fr (A.S.); pascal.richomme@univ-angers.fr (P.R.); 4Institute of Pharmaceutical Biology and Phytochemistry, University of Münster, Corrensstraße 48, D-48149 Münster, Germany; ahensel@uni-muenster.de

**Keywords:** mycosporine-like amino acids, stereochemistry, X-ray crystallography, ECD, anti-aging, wound healing

## Abstract

Mycosporine-like amino acids (MAAs) are water-soluble metabolites, reported to exhibit strong UV-absorbing properties. They have been found in a wide range of marine organisms, especially those that are exposed to extreme levels of sunlight, to protect them against solar radiation. In the present study, the absolute configuration of 14 mycosporine-like-amino acids was determined by combining the results of electronic circular dichroism (ECD) experiments and that of advanced Marfey’s method using LC-MS. The crystal structure of a shinorine hydrate was determined from single crystal X-ray diffraction data and its absolute configuration was established from anomalous-dispersion effects. Furthermore, the anti-aging and wound-healing properties of these metabolites were evaluated in three different assays namely the inhibition of collagenase, inhibition of advanced glycation end products (AGEs) and wound healing assay (scratch assay).

## 1. Introduction

Mycosporine like amino acids (MAAs) are water-soluble secondary metabolites mainly produced by marine organisms and less often by freshwater or terrestrial species [1]. They are composed of a cyclohexenimine or cyclohexenone skeleton, conjugated with amino acids or with imino alcohol substituent groups [2]. They are well known for their high absorption coefficients ranging from 10,000–50,000 M^−1^·cm^−1^ [3] and their UV-absorbing properties, which are responsible for the high attention of the scientific community and cosmetic industry [1]. MAAs are proposed to be biosynthesized via two possible biosynthetic ways, either the shikimate pathway [4] or another route which involves sedoheptulose 7-phosphate; the latter is a product of the pentose phosphate pathway [5]. In both pathways, 4-deoxygadusol is the precursor for all MAAs.

In the past, a limited number of studies addressed the stereochemistry of specific MAAs, and those presenting experimental data concluded that the ring stereogenic center is in the *S* form. In 1980 Furusaki et al. [6] performed the X-ray crystal analysis of palythine, reporting that palythine existed as zwitter ion and that palythine and water molecules were connected through hydrogen bonds, however, the absolute configuration of the C-1 was not described; Furthermore, Uemura et al. [7] performed the X-Ray crystal analysis of palythene showing that the absolute configuration of the C-1 was *S*. White et al. in 1995 performed the asymmetric synthesis of mycosporine serinol and mycosporine glycine concluding that the absolute configuration of the ring stereogenic center in both molecules was in the *S* form, too [8]. Finally, Klisch et al. [9] performed an experimental and calculational NMR investigation on the stereochemistry of porphyra-334 in 2007, and they stated that the absolute configuration of the stereogenic center of the ring was *S*. In contrast, more recent studies indicated the presence of MAAs with *R* configuration at the same position, although detailed explanations are missing [10,11]. Remarkably, the stereogenic center can be changed from *S* to *R*, due to changes of the priority number of substituents due to decarboxylation [3]. The reason why stereochemistry is so important is its possible influence on bioactivity. There are many examples of enantiomeric compounds demonstrating differential biological effects, underlying the significance of absolute configuration determination [12,13,14]. Thus, in order to study the pharmacological effects of MAAs meaningfully, their stereochemistry also needs to be established.

The production of MAAs may serve diverse purposes in the organism. The majority of studies are focussing on their photo-protective and antioxidant properties, but MAAs are also reported to protect cells against salt, desiccation, and thermal stress and can act as intracellular nitrogen storage and accessory pigments in photosynthesis [3,15]. As a consequence a plethora of biological and cosmeceutical properties of MAAs for humans have been studied, mainly focussing on prevention of the harmful effects of UV radiation on the human body. In recent studies different cell models, (human skin fibroblasts and HaCaT keratinocytes) were used to investigate the UV protective effects of the most common MAAs, namely porphyra-334, shinorine, palythine, and mycosporine-glycine to prove their efficacy as potential sunscreens [1] and references therein. Additionally, several anti-aging activities have been reported, especially targeting on the elastic fibers of the extracellular matrix (ECM) such as collagen, elastin and their remodeling enzymes [16,17,18,19,20]. Additionally, two marketed products are available, both composed of an extract with enriched or defined MAA concentration under the name of Helioguard^®^365 and Helionori^®^ [21,22].

The focus of the present study was to determine the absolute configuration of previously isolated MAAs shown in Figure 1, and to examine whether they possess in vitro anti-aging and wound-healing properties. For this purpose, thirteen MAAs were investigated in three different assays for their ability to inhibit collagenase and advanced glycation end products (AGEs) and in a cell-based scratch assay to evaluate their potential wound healing properties by influencing human keratinocyte cell physiology.

## 2. Results 

### 2.1. Determination of the Absolute Configurations of Amino Acids in MAAs by the Advanced Marfey’s Method Using LC-MS

Advanced Marfey’s method for compounds **1**–**5** as well as for compounds **8**–**12** and **15** [23] showed in all cases the presence of *L*-amino acids and *R*-threamine as the constituent of the final products of the reaction (Appendix A) indicating that the absolute configuration of all amino acids at the studied MAAs was *L* configuration.

### 2.2. Determination of Absolute Configuration of Isolated MAAs by Quantum Chemical Calculation Method

In order to determine the absolute configuration in chiral center 1, ECD calculations were carried out for six compounds (**1**, **3**, **7**, **9**, **10** and **11**), and results were then extended to all the others based on similarity of the structures and a comparison of the experimental ECD spectra with the calculated ones. For example, the results for compound **4** were compared to those of compound **1**, because only the serine group in compound **1** was exchanged by alanine in compound **4**; compounds **2** and **12** were compared to **3** as only the side chain was prolonged without any additional chiral center; compound **15** was matched with **9** as both also only differed in the structure of the side chain (glutamine versus glutamine acid), which should not have any effect on the ECD spectrum due to the distance from the chromophore; compound **8** to compound **10** due to the similarity of side chains; and compound **11** was calculated separately as it possesses an additional chiral group in the side chain and thus showed no similarity with the other investigated compounds; compounds **5**, **6**, and **14** were compared to **7**, because similarity arises from the presence of a glycine moiety on C-5 and non-amino acid groups (with no chromophore) in the second side chain.

The 3D structures of the compounds were drawn with *S* or *R* configuration in C-1 and then subjected to conformational analysis using MacroModel 09 (Schrödinger Ltd.) with OPLS-3 as force field in water and by implementation of Monte Carlo method. The number of steps were set sufficiently high to include all low energy conformers. Conformers occurring in an energy window of 2 Kcal.mol^-1^ were further subjected to geometrical optimization using wb97xd/6-31+g(d,p)/SMD in water (except for compound **11**, which was optimized with wb97xd/6-31g(d,p)/SMD/water). The Boltzmann averaging of the minimized conformers is shown in Appendix A. Subsequent simulation of electronic circular dichroism of the compounds was carried out using TD-DFT/M062x/6-31G+(d,p)/SMD/water. Selection of these parameters was due the flexibility of the molecules and their side chains, for which the implemented method showed to be suitable for this purpose [24]. As shown in Figure 2 and Appendix A, ECD spectra of all compounds revealed a negative cotton effect (CE) around 225 nm and a positive cotton effect around 200 nm, and all simulated spectra were in good agreement with experimentally recorded ones. Small deviations between calculated ECD spectra are due to the overestimation of calculation method and the contribution of different conformers in the solution, which could possibly be removed by computational methods. ECD spectra of all compounds were measured in water and compared to the calculated spectra as shown in Figure 2. Although this approach is sufficient for determining the absolute configuration of all compounds, measured optical rotation of some of the compounds was also taken into consideration. Therefore, the respective values of compounds **1** and **4**; **2**, **3**, and **12**; as well as **6**, **7**, and **14** were compared. As in each of the three groups no extra chiral center was present in the molecules, it is expected that the prefix of optical rotation (sign) must remain the same, which was in accordance with our observation (negative values for all MAAs). Thus, the absolute configuration of compounds **1**–**4**, **8**, **10**–**12** was determined as *S* and of the compounds **5**–**7**, **9** and **14**–**15** determined as *R*.

The results of this computational study are also in agreement with those of the crystal structure determination for compound **1**, which is discussed in the following section.

### 2.3. Crystal Structure of the Shinorine Hydrate 1H

Single crystals were obtained by slow evaporation from an aqueous solution of shinorine. The crystal structure determination showed the presence of a hydrate with the composition 1 · 1.72 (H_2_O). The corresponding crystallographic data and details of the structure refinement are collected in Appendix A. The asymmetric unit consists of two shinorine molecules (denoted as molecules *A* and *B*, see Figure 3), one fully occupied water position (*w1*) and a second water molecule *w2* which is disordered over two positions (occupancy ratio 0.7:0.3). There are also three partially occupied water positions, collectively denoted as *w3*, where the major disorder component is defined by a single water position (occupancy 0.56), while the other two positions are simultaneously occupied in the minor component (occupancy 0.44). The overall shinorine/water ratio in the crystal is therefore 1:1.72. In molecule B, the ‒CH_2_‒OH substituent bound to C1B is disordered over two alternative conformations (occupancy ratio 0.79:0.21). Both independent shinorine (**1**) molecules exist in a zwitterionic state as depicted in Figure 1. The chirality at the C1A and C1B centers of molecules A and B (Figure 3) is *S*, and the chirality at both C11A and C11B is also *S*.

Each shinorine molecule of type *A* is bonded to a *B*-type molecule via four N‒H···O bonds (Figure 4a), and two dimeric units of this kind are bridged by a *w1* water molecule via (*A*)O‒H···O(*w1*)···H‒O(*B*) interactions. Altogether, shinorine (*A* + *B*) and *w1* water molecules form an extensive network based on strong H-bond interactions (Figure 4b). This framework displays large open channels propagating parallel to the crystallographic *a* axis. These channels are occupied by disordered *w2* and *w3* water molecules which are H-bonded to the main framework.

### 2.4. Collagenase Inhibition Assay

All MAAs that were available in a larger amount were assessed in vitro by a spectrofluorometric method for their collagenase inhibitory activity. The effectiveness of the studied compounds was evaluated in the concentration range of 1 to 125 µg/mL (1 to 350 µM). All test results are summarized in Table 1 and the respective concentration response curves can be found in Appendix A. Compound **9** (58.39 µM) showed the strongest inhibitory effect followed by **6** (70.91 µM), **13** (80.52 µM), and **5** (80.71 µM). Compounds **11** and **7** showed only moderate activity with an IC_50_ around 160 µM. **4** was notably less effective than the others. In many publications EGCG is used as a positive control and published with IC_50_ values around 220 µM [25]. At these concentrations EGCG shows a fluorescence quenching effect and does not give reliable inhibition values. This effect is no longer present at lower concentrations; therefore a dose dependent inhibition of EGCG with an IC_50_ of 46.62 µM could be measured. In addition, rutin was evaluated as a standard inhibitor as well, which resulted in an inhibition of 52% at 100 µM. In comparison to other pure compounds with reported collagenase inhibition, such as quercetin-glycosides (30% to 40% inhibition at 100 µg/mL) [26], taxifolin (IC_50_ 193.3 μM), *E*-piceid (258.7 μM), *E*-astringin (124.9 μM), or taxifolin 3′-*O*-glucopyranoside (141.4 μM) [27], the majority of MAAs show a significant inhibition of collagenase.

### 2.5. Advanced Glycation End Products (AGEs) Assay

In total 10 MAAs were investigated for their anti-AGEs activity. The substances were evaluated in the concentration range of 1 µM to 3 mM. All MAAs showed a dose-dependent inhibition and their IC_50_ values are summarized in Table 1 and dose-effect curves are can be found in Appendix A. All tested compounds showed lower IC_50_ values than aminoguanidine (1.4 mM), usually used as a positive control. The most active compounds, i.e., **4** (IC_50_ = 75 µM), **10** (85 µM), **2** (90 µM), and **1** (103 µM) exhibited comparable anti-AGEs effect as rutin (85 µM), a quercetin glycoside. Used as a positive control, the aglycone part, i.e., quercetin, was previously described as a strong AGEs inhibitor [28] such as different flavonoids (50–90 µM) [29]. Indeed, using the same assay, many antioxidant polyphenols inhibited AGEs formation, probably mainly through their antioxidant potential [30,31]. Some secondary derivatives isolated from lichen (variolaric acid, pannaric acid and leprapinic acid) by Schinkovitz et al. [32] were also recently described with similar IC_50_ values between 50–80 µM but their activity could not be correlated to their radical scavenging activity. Indeed, the formation of AGEs may be prevented by an antioxidant but also by 1,2-dicarbonyl scavengers [29] or AGEs breakers [33]. As far as MAAs are concerned, their structure make them prone to trap carbonyl species. 

### 2.6. Influence of MAAs on Migratrion Behaviour of Human Keratinocytes (Scratch Assay)

Based on the results of the two enzyme assays shown previously, 4 MAAs that had been isolated in a larger amount were tested in a cell based scratch assay on human keratinocytes to evaluate potential wound healing activity. Compounds **1**, **2**, **4**, and **8** that had shown good results in both in vitro assays were investigated in two concentrations of 1 µM and 10 µM on the non-tumorgenic, spontaneous immortalized human keratinocytes cell line HaCaT. All the treated groups showed significant migration and narrowing of the scratch area after 24 h (*p* < 0.05). The power of cell migration was comparable for each compound, with bostrychine B showing a slightly higher wound closure ability. All the groups treated with different MAAs showed significant closure of the scratch area, comparable to the positive control group. The calculated % of wound closure for each experiment is displayed in Figure 5. The data obtained by the scratch assay indicate that MAAs can interact with the cell physiology of human keratinocytes, leading to increased proliferation and migration towards an enhanced wound closure ability. The actual mechanisms of action behind this observed activity are however unclear and further experiments needs to be carried out.

## 3. Discussion and Conclusions

Within the present study, the absolute configuration of 14 MAAs was determined using a combination of Marfey’s reaction and ECD calculations. Furthermore, the X-ray crystallographic analysis of shinorine was performed for the first time. Chirality at the stereogenic center at C1 of the cyclohexenimine ring always proved to be *S*, and it was shown that the shinorine molecules form an extensive network of intramolecular strong H-bonds with one another and with water molecules.

Our data are in agreement with previously published data for porphyra-334, which suggested that the configuration of the stereocenter C-1 was *S* [9]. Additionally, our results showed that the configuration of the stereocenter C-1 in shinorine and palythine was *S* and *R*, respectively which can be explained by changes of the priority number of substituents of the side chains.

Understanding the correct stereochemical configuration of the MAAs is a prerequisite for understanding functionality and docking into biological systems and when it comes to pharmacological investigations those delicate molecular differences can have great impact on the respective activity of a single compound. Accordingly, by unambiguously clarifying this aspect for 14 representatives of this highly important class of natural products, our results significantly contribute to a better understanding of their biological role and possible use.

During aging the skin is exposed to various stressors such as UV radiation that can alter functionality, quality, and the character of the ECM [34]. Collagen, the major structural protein of the ECM provides a supportive framework to the cell and is responsible for tensible strength, elasticity, and hydration of the skin. In inflammed tissue after UV radiation or during wound healing an increased production of degrading dermal enzymes (elastase and collagenase) lead to the subsequent degradation of ECM. This leads to reduced functionality, elasticity, hydration, and thickness of the dermal connective tissue negative also influencing the correct formation, proliferation, and differentiation of keratinocytes from the basal layer towards an intact epidermal barrier. Typical symptomatic consequences such as wrinkle formation and sagging of the skin but also decelerated wound healing process is observed from the clinical point of view [35]. Our results show that all tested MAAs are able to inhibit collagenase in a dose dependent manner, and therefore MAAs could play a role in different phases of the wound healing process and extrinsic aging. These findings are in accordance to results from Ryu et al. [20], who describe a concentration-dependent inhibition of UV-A induced MMPs such as MMP-1 and elastase, and an increase in procollagen production for porphyra-334 in concentrations at 40 µM. Regarding a possible structure–activity relationship, bostrychine-C, which is substituted with glutamic acid only in position 5 and carries no other amino acid, showed the highest inhibition of collagenase. In addition, asterina-330 and aplysiapalythine A, carrying a glycine residue and only a small functional group such as glycinol (asterina-330) in position 3, showed a good inhibition, too. Therefore, medium size MAAs seemed to be more active compared to molecules with bigger side chains in both positions such as bostrychines D and E. MAAs carrying both, a glycine and a threonine-like side chain were all medium active. In contrary, less polar side chains such as methylamine (mycosporine–methylamine–threonine) seemed to have a negative effect on activity.

Twelve MAAs based on a cyclohexenimine skeleton and only one MAA with cyclohexenone substructure were tested. The cyclohexenone type mycosporine-glycine showed a very pronounced inhibition. However, as it was the only representative of this subtype it can only be speculated whether the carbonyl-group has a positive effect on the activity or not. Reliable structure–activity relations can only be predicted in molecular docking studies, which would be an interesting topic for further studies. The mechanism of collagenase inhibition by MAAs still remains unknown; however, there is the hypothesis that the metal chelating activity of MAAs could play a role due to the chelation of iron and calcium ions [17,36].

In addition, advanced glycation end products (AGEs) formation is a natural occurring process in normal chronical aging. High AGEs levels have been associated with the pathogenesis of various diseases and recently AGEs have received particular attention in skin aging and wound healing research [37]. One of the major target structures for glycation is collagen and once carbohydrate binds irreversible to it, the collagen fiber loses its elasticity and becomes shorter which may result in delayed wound healing, too [38]. Our results show that MAAs can prevent from advanced glycation with IC_50_ values around 75 to 90 µM. The respective activity is comparable to the positive control rutin (80 µM) and much lower than that of the standard compound aminoguanidine. These results are the first evidence that MAAs show a beneficial effect in preventing the formation of AGEs. Recently mycosporine-2-glycine was found to inhibit the glycation-dependent cross-linking of hen egg white lysozyme (HEWL) with an IC_50_ of 1.61 mM, which is another indication for the anti-AGEs activity of MAAs. MAAs that contain threonine or other amino acids with similar size at position 3 as well as an additional functional group with a carboxylic acid in the other side chain showed the highest activities. On the contrary, palythine with only one acidic group and no side chain in position 3, showed the weakest effect. Further studies are definitely needed to understand the importance of substitution patterns on the activity of MAAs, however their antioxidant properties might also be contributing factors by preventing oxidation of the Amadori product [39]. In addition, the beneficial effects of four MAAs on the main cell types of the epidermal layer, keratinocytes were confirmed in a cell based approach using the scratch assay model. Our results indicate that already low MAA concentrations of 1 and 10 µM accelerated the cellular proliferation and direct migration, congruent to the potential “healing process” after 24 h of treatment. Summarizing, the results of this study indicates that MAAs may be valuable ingredients in dermo-cosmetic formulations and moreover in wound healing formulations as well. Finally, only a few MAAs, namely shinorine, porphyra-334, palythine and mycosporine-2-glycine have been evaluated for their safety (toxicity) in HaCat cells and human keratinocytes, [40,41], in human lung fibroblasts (WI-38) as well as normal human skin fibroblasts (TIG-114) [20,42]. For dermo-cosmetic applications, however further studies are needed including the evaluation of other pure MAAs and extracts containing a defined MAA concentration in order to create safety profiles and avoid adverse effects.

## 4. Materials and Methods

### 4.1. Isolated Mycosporine-Like Amino Acids

Shinorine (compound **1**), porphyra-334 (compound **2**), mycosporine-methylamine-threonine (compound **3**), mycosporine-alanine-glycine (compound **4**), aplysiapalythine A (compound **5**), asterina-330 (compound **6**), palythine (compound **7**), mycosporine-glycine (compound **13**), and aplysiapalythine B (compound **14**) were isolated as described in [43] and bostrychine-A (compound **15**), bostrychine-B (compound **8**), bostrychine-C (compound **9**), bostrychine-D (compound **10**), bostrychine-E (compound **11**), and bostrychine-F (compound **12**) were isolated as described in [23] before and all measurements for the proof of identity and purity can be sent upon request.

### 4.2. Instrumentation

LC-MS data recorded following Marfey’s method were measured with an Agilent InfinityLab LC/MSD System (Santa Clara, CA, USA), comprising of an Agilent HPLC 1260 HPLC equipped with binary pump, autosampler, column oven, and photodiode array detector. 

### 4.3. Chemicals and Reagents

Solvents for analytical experiments in analytical quality (p.a.) were obtained from Merck (Darmstadt, Germany). Ultrapure water was produced by a Sartorius arium^®^ 611 UV (Göttingen, Germany) purification system.

### 4.4. Determination of the Absolute Configurations of Amino Acids in MAAs by the Advanced Marfey’s Method

The general procedure was adapted from the advanced Marfey’s method [44] and was the same as described by Orfanoudaki et al. in 2019 [23]. Approximately 0.1 mg of each MAA was stirred with 37% HCl (250 μL) at 90 °C for 120 min. The hydrolysates were evaporated to dryness and then resuspended in H_2_O (100 μL). 

Each amino acid (200 μg) or hydrolysate was added to 1 M NaHCO_3_ (200 μL) and 1% d- or l-FDLA in acetone (25 μL). The reaction vials were incubated and stirred for 30 min at 50 °C. The reactions were then quenched with 2 N HCl (100 μL). MeOH (800 μL) was added to prepare LC-MS samples. The reaction products were analyzed using a Luna 5u C-8 column (150 × 4.6 mm, 5 µm; Phenomenex, Torrance, CA, USA). H_2_O containing 0.1% formic acid (A) and MeCN containing 0.1% formic acid (B) were used as eluents. The solvent gradient was as follows: 0 min: 5% B, 1 min: 5% B, 20 min: 20% B, 40 min: 60% B, 45 min: 95% B, 45.1 min: 5% B and 55 min: 5% B. The DAD detector was set to 210, 254, 280, 320, 330, 350, and 400 nm, the flow rate, injection volume, and column temperature were adjusted to 0.9 mL/min, 20 μL, and 40 °C, respectively. MS spectra were recorded in positive-ESI mode (capillary voltage 4.5 kV), with a drying gas temperature of 300 °C, the nebulizer gas (nitrogen) set to 25 psi, and a nebulizer flow (nitrogen) of 12.0 L/min. The scanned mass range was set between *m/z* 50 and 600 (Appendix A).

### 4.5. Measurement of ECD Spectra and ECD and Optical Rotation Calculation

3D structures of isolated compounds were drawn in Maestro (Schrödinger. LLC, New York, NY, USA) and subjected to conformational analysis using MacroModel 9.1 (Schrödinger. LLC and OPLS-3 as force field in water by implementation of Monte Carlo method. Geometrical optimization and energy calculation of conformers occurring in an energy window of 2 Kcal·mol^−1^ were done by implementation of DFT/wb97xd/6-31+g(d,p)/SMD in water phase by using Gaussian 16 (Revision A.03, Gaussian, Wallingford, CT, USA 2016). In case of compound **11**, the same method used without diffusion parameter. Subsequently, ECD spectra of optimized compounds were simulated by using TD-DFT/M062x/6-31+G(d,p)/SMD/water. Obtained ECD spectra (with half-band of 0.25–0.3 eV and UV shift of 8–20 nm were Boltzmann averaged and scaled spectra compared with experimental spectra obtained in water.

### 4.6. X-ray Crystallography

Intensity data were collected at 193 K, using Cu radiation (λ = 1.54184 Å), on an Oxford Diffraction Gemini-R Ultra diffractometer operated by the CrysAlisPro software [45]. The data were corrected for absorption effects by means of comparison of equivalent reflections. The structure was solved with the direct methods procedure implemented in SHELXT [46] and refined by full-matrix least squares refinement on *F*^2^ using SHELXL-2014 [47]. Non-hydrogen atoms were located in difference maps and refined anisotropically. The hydrogen atoms bonded to C atoms, O13A and O13B were fixed in idealized positions. All other H atoms bonded to O and other H atoms bonded to N atoms were refined using distance restraints [O‒H = 0.84(1) Å and N‒H = 0.88(1) Å. Distance restraints for chemically equivalent 1,2- and 1,3-distances were used in the refinement of the disorder in molecule *B*. The absolute structure was established by anomalous-dispersion effects in diffraction measurements on the crystal and a Flack *x* parameter of 0.09 (14) was determined from 2233 quotients [(I^+^) − (I^‒^)]/[(I^+^) + (I^‒^)] [48].

### 4.7. Collagenase Inhibition Assay

The assay was carried out according to a previously published protocol by Hartmann et al. [18]. All required reagents for the assay were purchased from Sigma Aldrich. Both enzyme (collagenase type V; C-9263) and substrate (SCP0192 MMP-2 substrate, Sigma Aldrich) were dissolved in the reagent buffer (Tris–HCl; 10 mM pH 7.3) and diluted from stock solutions to 100 μg/mL (initial) enzyme concentration and 55.55 μg/mL (initial) substrate concentration prior to use. Test samples and standard inhibitor phosphoramidon disodium salt used as positive control were dissolved in H_2_O/DMSO, always maintaining a final DMSO concentration of 1% in each well. The final reaction volume of 100 μL was composed of 25 μL buffer solution, and 25 μL sample/positive controls which were added first followed by 25 μL enzyme. This mixture was incubated for 10 min at 37 °C. Finally, 25 µL of substrate were added to initiate the readout reaction. The substrate which carries a fluorogenic residue (7-methoxycoumarin-4-yl acetic acid) is continuously dissipated by the enzyme collagenase. Measurements are taken after 30 min at an excitation wavelength of λ = 320 nm and λ = 400 nm emission wavelengths in a 96-well microplate format using an Infinite F 200 pro microplate reader equipped with filter-based technology (Tecan, Männedorf, Switzerland). An untreated control group containing 1% DMSO without sample was set to 100% enzyme activity. Blank wells without enzyme were measured accordingly and subtracted from the fluorescent units of each well. Two positive controls were used. First, the well-established phosphoramidon disodium salt and second, to include natural product based inhibitors, rutin and EGCG were used. Measurements were performed in triplicates using 96 well plates, and each substance was measured in three independent experiments. Collagenase inhibition is expressed as
(1)$$\%\;\mathrm{Inhibition}\;\mathrm{Collagenase}\;=\left[1-\frac{\left(A-B\right)}{\left(C-D\right)}\right]\times 100$$
where A indicates the absorbance of reaction mixture containing both enzyme and sample, B shows the absorbance of the reaction mixture without enzyme, C stands for absorbance of reaction mixture with enzyme only and D marked for the absorbance of reaction solution without enzyme and sample. IC_50_ values were calculated by fitting data (log (Inhibitor) vs. response–variable slope) using GraphPad Prism (GraphPad Software, San Diego, CA, USA).

Statistics: The data were analyzed at least in three independent experiments and were expressed as mean ± SEM. A Shapiro Wilk normality test was performed to confirm the Gaussian distribution of the data. One-way ANOVA, followed by a Dunett’s multiple comparison test in GraphpadPrism 5 were performed to clarify statistical significance. *p*-Value of ≤ 0.05 was considered statistically significant.

### 4.8. Inhibition of Anti-Advanced Glycation End Products (AGEs) 

The anti-AGE assay was carried out following the methodology previously reported by Séro et al. [28]. Briefly, the samples (1 µM to 3 mM) were incubated with D-ribose (0.5 M) and bovine serum albumin (BSA, 10 mg/mL) in a Na-phosphate buffer (50 mM pH 7.4); the solutions were incubated in black microtiter plates (96 wells) at 37 °C for 24 h in a closed system before AGE fluorescence measurement. Both vesperlysines-like (λ_exc_370 nm; λ_em_440 nm) and pentosidine-like (λ_exc_335 nm; λ_em_385 nm) AGE fluorescence were measured using a microplate spectrofluorometer Infinite M200 (Tecan, Lyon, France). To avoid quenching phenomena, the fluorescence resulting from the incubation in the same conditions of BSA (10 mg/mL) and the tested extract (1 µM to 3 mM) was subtracted from each measurement. The negative control, i.e., 100% inhibition of AGEs formation, consisted of wells with only BSA. A control, i.e., no inhibition of AGEs formation, consisted of wells with BSA (10 mg/mL) and d-ribose (0.5 M). The final volume assay was 100 µL. The percentage of AGE formation was calculated as follows for each extract concentration: AGEs (%) = (fluorescence intensity [sample]−fluorescence [blank of sample]/fluorescence intensity [control]−fluorescence [blank of control]) × 100.

Dose-effect curves were best fitted with a sigmoidal dose-response equation [four parameters logistic model: bottom and top plateaus, the EC_50_, and the slope factor (Hill’s slope)] using Sigma Plot 14.0 software, which enabled calculation of the IC_50_ values. The acceptance criteria were established by the correlation coefficient (r²) > 0.97. Results were compared with those of reference products, aminoguanidine and rutin.

### 4.9. Wound Scratch Assay

HaCaT keratinocytes were provided by Prof. Dr. Norbert E. Fusenig (DKFZ, Heidelberg, Germany) and cultivated in a DMEM medium, supplemented with FCS (10%), penicillin/streptomycin solution (1%), and non-essential amino acids (1%) (PAA, Pasching, Austria). HaCaT keratinocytes were cultivated at 37 °C, 5% CO_2_. 

The evaluation of 4 different MAAs (compounds **1**, **2**, **4**, and **8**) on the migration of HaCaT cells was determined as described by Balekar et al. [49]. HaCaT cells were plated into a 12-well plate at a concentration of 1 × 10^5^ cells containing Dulbecco’s modified Eagle’s medium culture medium (DMEM) supplemented with 10% fetal calf serum (FCS) and incubated at 37 °C in a humidified 5% CO_2_ atmosphere until confluence has been reached. After attachment of the cells to the plate, medium was removed and the adherent cell layer was scratched with a sterile pipette tip. Cellular debris was removed by washing off with PBS (2 × 1 mL). Subsequently, the cells were treated with DMEM supplemented with 10% FCS containing the individual MAA test compounds at the concentrations of (10 µM to 1 µM). Control cells received only fresh DMEM +10% FCS, cells used as the positive control were treated with fresh DMEM + 20% FCS. Following, the cell cultures were incubated at °C in a humidified 5% CO_2_ atmosphere. Finally, the area of each scratch was determined by quantitative imaging at time 0 (just after scratching cell monolayers) and after 24 h treatment with the compounds (Leica DFC300, Wetzlar Germany; 40× magnification). The captured images were then analyzed using Image J software [50].

The migration rate can be expressed as the percentage of area reduction or wound closure:(2)$$\mathrm{Wound}\;\mathrm{closure}\;\%=100\;\left[\frac{\mathrm{Area}\;\left(t0\right)-\mathrm{Area}\left(\;t24\right)}{\mathrm{Area}\;\left(t0\right)}\right]$$

Statistical analysis: The data were analyzed in three independent experiments and were expressed as mean ± SEM. One-way ANOVA, followed by an unpaired t-test to compare control group and treated group was performed. *P*-value of ≤ 0.05 was considered as statistically significant.

## Figures and Tables

**Figure 1 marinedrugs-18-00035-f001:**
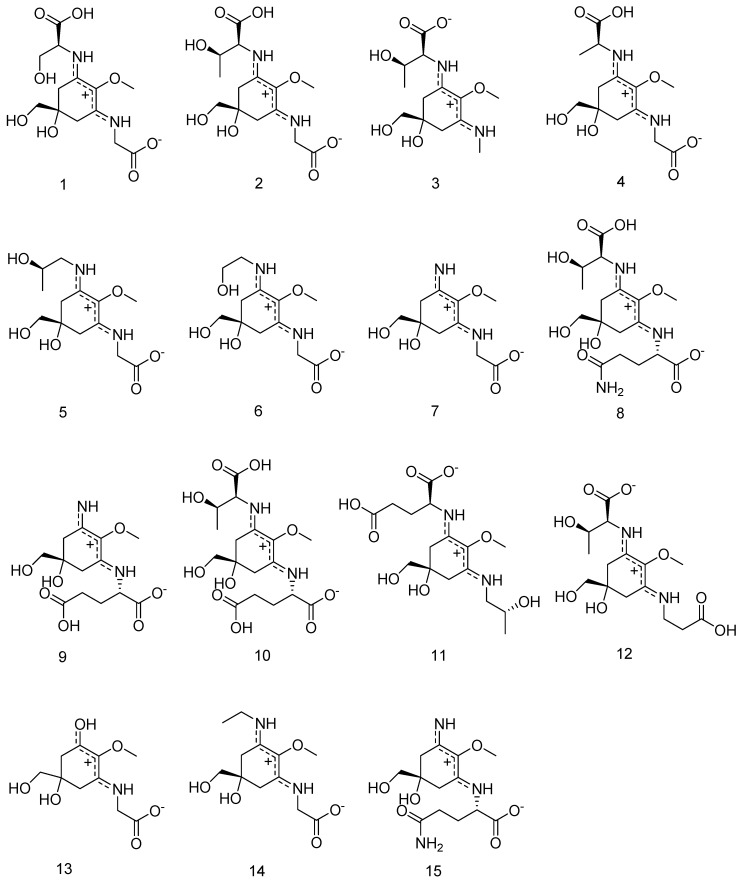
Structural features of selected Mycosporine-like amino acids (MAAs), integrated into the studies.

**Figure 2 marinedrugs-18-00035-f002:**
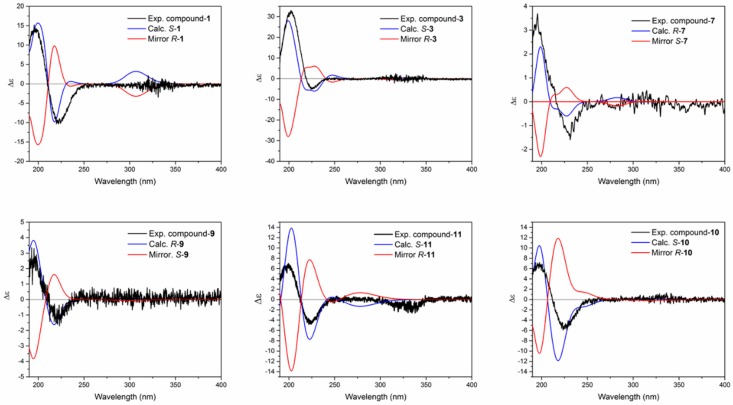
Experimental vs. calculated electronic circular dichroism (ECD) spectra of compound **1**, **3**, **7**, **9**, **10** and **11.**

**Figure 3 marinedrugs-18-00035-f003:**
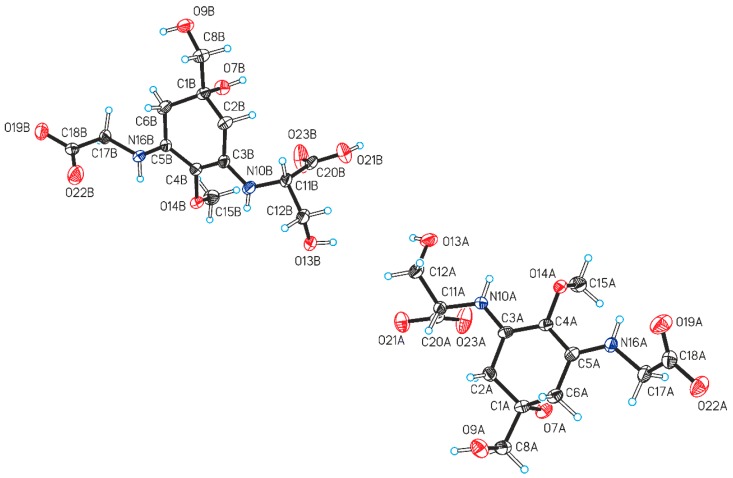
Two independent shinorine molecules in the crystal structure of 1H with thermal ellipsoids of non-H atoms drawn at the 50% probability level and H atoms represented as spheres of arbitrary size. The minor disorder component in molecule *B* has been omitted for clarity.

**Figure 4 marinedrugs-18-00035-f004:**
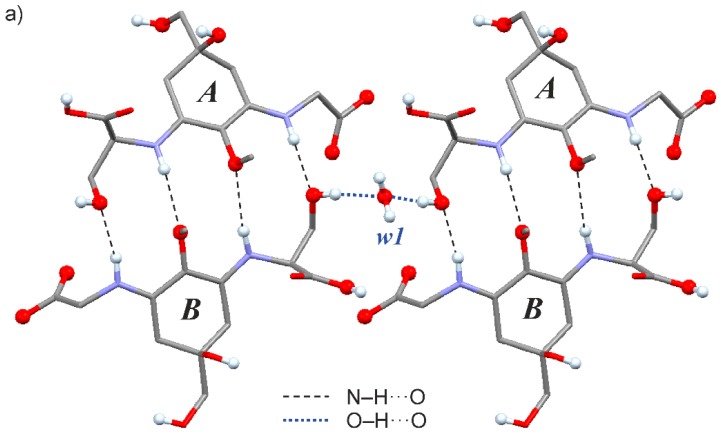
Detail of the H-bonded structure of 1H. (**a**) Two sets of dimers of four-fold N‒H···O bonded pairs of molecules *A* and *B*. A central water molecule *w1* bridges the two dimeric units via (*A*)O‒H···O(*w1*)···H‒O(*B*) interactions. (**b**) N‒H···O and O‒H···O bonded network formed by shinorine *w1* water molecules. The disordered water molecules *w2*, *w3* (not shown) occupy the central cavity. O and H atoms directly engaged in any intermolecular H-bond interaction are drawn as balls.

**Figure 5 marinedrugs-18-00035-f005:**
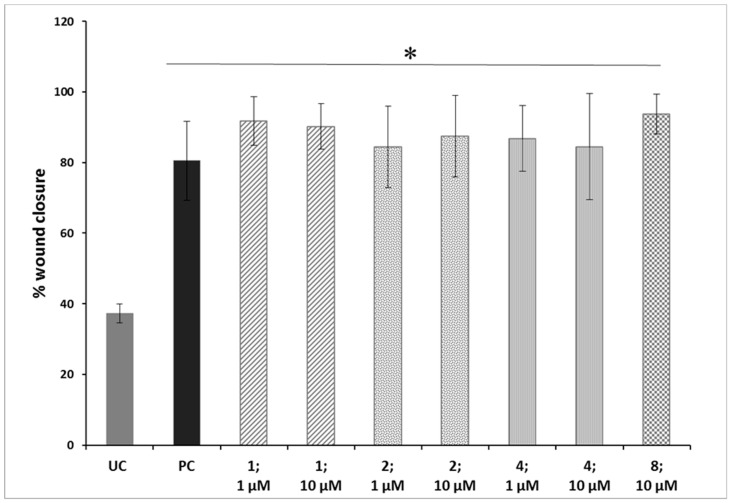
Influence of different Mycosporine-like amino acids on the relative ability (%) of wound closure after 24 h with scratch assay on HaCaT cell line. UC: untreated control; PC positive control medium supplemented with 20% fetal calf serum FCS); 1: shinorine; 2: porphyra-334; 4: mycosporine-glycine-alanine; 8: bostrychine B; * *p* < 0.05 in comparison to UC. Values are based on *n* = 3 independent experiments.

**Table 1 marinedrugs-18-00035-t001:** Summary of all compounds and their IC_50_ values in the two enzyme assays collagenase inhibition and advanced glycation end products (AGEs) inhibition. IC_50_ values with corresponding 95% confidence intervals (CI_95_).

Compound	Collagenase Inhibition	Pentosidine-Like AGEs
	IC_50_ µM (CI 95 ±)	IC_50_ µM
Compound **1** ^+^	104.0 (95.54 to 110.8)	103
Compound **2** ^+^	105.9 (94.43 to 117.8)	90
Compound **3**	250.5 (237.4 to 264.3)	150
Compound **4**	158.1 (153.0 to 163.4)	75
Compound **5**	80.71 (73.29 to 88.89)	400
Compound **6**	70.91 (65.53 to 76.73)	125
Compound **7** ^+^	158.9 (141.4 to 175.7)	700
Compound **8**	104.5 (98.51 to 111.0)	-
Compound **9**	58.39 (55.51 to 61.41)	-
Compound **10**	118.0 (108.1 to 128.7)	85
Compound **11**	163.0 (150.6 to 176.5)	150
Compound **12**	90.03 (82.18 to 98.64)	200
Compound **13**	80.52 (74.56 to 86.96)	-
Phosphoramidon **	1.90 (1.765 to 2.039)	-
Epigallocatechingalleate **	46.62 (41.30 to 52.63)	-
Rutin **	100 µM -> 52% Inhibition *	85 µM
Aminoguanidine **	-	1.4 mM

* 1% DMSO control group was set to 100% enzyme activity; ** used as positive control; ^+^ Collagenase results previously published in Hartmann et al. [18].

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
