# Peer review of "Absolute Configuration of Mycosporine-Like Amino Acids, Their Wound Healing Properties and In Vitro Anti-Aging Effects"

_marinedrugs, 2019, doi:10.3390/md18010035_

Round 1

Reviewer 1 Report

The manuscript "Absolute Configuration of Mycosporine-Like Amino Acids, their Wound Healing Properties and in vitro Anti-ageing Effects" by Hartmann and coll. is clear and well written. The study of the properties and absolute configuration of MAAs is very interesting. Unfortunately, the part on absolute configuration determination is not convincing.

The referee agrees with the absolute configuration determination of compound 1 by X-ray, but the use of ECD spectroscopy and moreover calculation of specific rotation are not valid:

- ECD signal is limited to one negative and weak band around 220 nm.

- ECD signal for compound 7 is too weak and very noisy.

- theoretical spectra should not be shifted by 40 nm, as written on line 303.

- if the dimers are strong, calculations should be done on dimers and not on monomers.

- the scales of figure 3 left and right are different (delta epsilon and delta theta), but the values are the same.

- figure S7 is not clear at all. Some geometries and Boltzmann populations are identical.

- line 105-106: the cotton is the same for compounds 1 and 7. The authors write the opposite.

- the reasoning on line 110-112 "As the position of CE are the same in all experimentally measured spectra of the compounds, one might conclude that the addition of the chiral center in the side chain has a minimum or no effect on ECD spectra" is particularly questionable.

- calculation of the sign of the specific rotation with polar=optrot in Gaussian is not a valid and efficient method to determine absolute configuration.

All the discussion about ECD spectroscopy must be removed, because the discussion does not have scientific value. The absolute configurations are not proved, except for compound 1.

Author Response

Thank you very much for your valuable comments which really helped to improve our manuscript.

The computational part was changed, and a new method was applied, which led to lower UV shifts of maximum +20 nm (approximately 10 nm for most of them). ECD spectra were re-measured and the CE at about 200 nm, which was much more intense than the 225 nm band, was also considered. Although, in the previous calculation, sodium D line wavelength has also been taken into account, the optical rotation calculation part was totally removed, and we extended the measured optical rotations based on the similarity of the compounds. Finally, regarding to the Boltzmann averaging part, we have added clear images of the conformers by including their superimposed Figures.

We tried our best to improve the manuscript and perform new calculations. However, if it is necessary we are going to perform calculations for compounds 8 and 10 in the future and make any other suggested change.

Reviewer 2 Report

The study performed by Peng He et al. focused on Absolute Configuration of Mycosporine-Like Amino Acids, their Wound Healing Properties and in vitro Anti-ageing Effects.

Overall the study is well conducted, and the objective of the study is interesting; however, I have some minor suggestions.
1/- IC50, complete with all details related to validation criteria of the fitted model.
Support the choice of concentrations in dose–effect curves because, for some products, the maximum effect is not reached,
Add graphical representation of the dose-response model for Pentosidine-like AGEs.
2/-Add all necessaries available details about LC-MS and LC-UV assay: mobile phase, columns, detection parameters, capabilities of the method LOD, LOQ, precision and matrix effect.
3/- Before applying parametric tests, the Gaussian distribution of data must be assessed. Is it the case? If not, use the non-parametric test.
4/- In the discussion, propose the mechanisms of action that explain the anti-ageing and wound-healing properties, as well as the precautions to be taken regarding adverse reactions and possible toxic effects to be tested in perspective.

Author Response

Thank you very much for your valuable comments which really helped to improve our manuscript.

1) The anti-AGEs assay was developed as a screening assay (Derbré et al. Anal. Bioanal. Chem. 2010 and Séro et al. Molecules 2013). It should be kept in mind that any associated measurement refers to a slow, nonenzymatic degradation process. In this kind of “chemical” assay, IC50s are thus much higher (mM) than the ones generally founded (µM) in “biological” assays (cytotoxicity, antifungal activity, antiparasitic activity…). When this assay was developed about ten years ago, every known synthetic or natural compound exhibited an IC50 in the millimolar range. Expecting better activities (µM), concentrations were chosen as detailled in the manuscript (1 µM to 3 mM).

As described in the papers cited above and in the experimental part: « A control, i.e. no inhibition of AGEs formation, consisted of wells with BSA (10 mg/ml) and D-ribose (0.5 M). The final volume assay was 100 µL. The percentage of AGE formation was calculated as follows for each extract concentration: AGEs (%) = (fluorescence intensity [sample] − fluorescence [blank of sample]/fluorescence intensity [control] − fluorescence [blank of control]) ×100. » It may explain why, for some products, the maximum effect was not reached.

Furthermore, graphical representations of the dose-response model for Pentosidine-like AGEs were present in the supporting information but they were replaced with a new Figure (Figure S9).

2) All details that were missing regarding the analytical analysis (LC-MS and LC-UV) were added to the text.

3) The datasets for each compound were assed as suggested to evaluate the Gaussian distribution of the data. Therefore, the Shapiro Wilk Normality test was performed in Graph Pad prism (see comment in section 4.7). For all data a Gaussian distribution could be confirmed.

4) Thank you for your comment. Structure activity relationships and discussion about the underlying mechanism that might explain the anti-aging and wound healing properties were added to the discussion. In addition, as suggested the literature about already existing toxicity studies on MAAs and a discussion on future investigations to ensure their safety and reduce adverse effects has been added.

Reviewer 3 Report

In this work absolute configuration of mycosporine-like amino acids is described. Also authors investigated their wound healing properties and in vitro anti-ageing effects. The absolute configuration of 14 mycosporine-like-amino acids was determined by combining the results of electronic circular dichroism (ECD) experiments and that of advanced Marfey’s method using LC–MS. The work is of interest because the results of this study indicates that MAAs may be valuable ingredients in dermo-cosmetic formulations and moreover in wound healing formulations as well. The article looks like a short communication and may be published after minor revision.

Notes:

Why refer to Figure 1 is followed after refers to Figures 2 and 3 in the text? The descriptions to ECD spectra on the Figure 3 especially right drawing should be increased. I think that Table 1 should be transferred to Supplementary Information. Why the evaluation of potential wound healing activity was not carried out for compound 8 in concentration of 1 μM? (Figure 6)

Author Response

Dear Reviewer 3:

Thank you very much for your valuable comments.

-Figures 1, 2 and 3 were placed in the right order (order mentioned in the text).

- The descriptions to the ECD spectra, (especially right drawing) were increased.

-Table 1 was transferred to the Supplementary Information.

- You are right with your comment about not testing compound 8 in lower concentrations. It would make sense to test this compound in lower concentrations as well.  Those experiments were carried out by one of the authors in course of a short term secondment to the laboratory in Münster, Germany. Unfortunately we didn’t have time to test Compound 8 in a lower concentration.

Reviewer 4 Report

The manuscript entitled “Absolute Configuration of Mycosporine-Like Amino Acids, their Wound Healing Properties and in vitro Anti-ageing Effects” by Orfanoudaki et al. describes the absolute configuration of 14 mycosporine-like-amino acids (MAAs). The anti-aging and wound-healing properties of some of these MAAs were also evaluated using collagenase inhibition assay, advanced glycation end products (AGEs) inhibition assay, and scratch assay. This is doubtlessly an interesting article that could be considered for publication. However, there are a few minor issues that should be addressed before its acceptance for publication.

The detailed comments are as follows:

Please explain why compounds 8, 9, and 13 are not included in the AGE inhibition assay, especially when compound 9 displayed very low IC50 in the collagenase inhibition assay. It is highly recommended that the authors should comment on the structure-activity relationship of these MAAs. The authors should perform an in vitro cytotoxicity study to determine the safety profiles of different MAAs.

Author Response

Dear Reviewer 4:

Thank you very much for your valuable comments which helped to improve our manuscript.

-Compounds 8, 9 and 13 were isolated in a small amount and were used for the evaluation of their collagenase inhibitory activity. Unfortunately, we didn’t have enough material left to investigate those compounds in both assays. But since it is the first report on MAAs and their ability to inhibit AGEs we still think that the 10 MAAs that were investigated already give quite a good overview on their activity in this assay.  We added a sentence to the paragraph 2.5 explaining why those compounds weren’t tested.

-For both Collagenase Inhibition and Anti-AGEs activity we added a discussion about their structure activity relationships for those two targets.

-Thank you for your comment. Given the narrow timeframe to answer to the comments (5 days) we were not able to perform any further experiments. However the literature about already existing toxicity studies on MAAs and a discussion on future investigations to ensure their safety and to reduce adverse effects has been added to the discussion.

Round 2

Reviewer 1 Report

The referee has noted all the changes made on the part on absolute configuration determination : the computational part was changed, and a new method was applied, which led to lower UV shifts . ECD spectra were shown from 190 nm and the CE at about 200 nm, which was much more intense than the 225 nm band, was also considered. The optical rotation calculation part was totally removed and clear images of the conformers have been added. This part has been greatly improved and is publishable.

The missing details in the other parts have also been added, so themanuscript "Absolute Configuration of Mycosporine-Like Amino Acids, their Wound Healing Properties and in vitro Anti-ageing Effects" by Hartmann and coll. could be considered for publication.